# Profiling Somatosensory Impairment after Stroke: Characterizing Common “Fingerprints” of Impairment Using Unsupervised Machine Learning-Based Cluster Analysis of Quantitative Measures of the Upper Limb

**DOI:** 10.3390/brainsci13091253

**Published:** 2023-08-28

**Authors:** Isuru Senadheera, Beverley C. Larssen, Yvonne Y. K. Mak-Yuen, Sarah Steinfort, Leeanne M. Carey, Damminda Alahakoon

**Affiliations:** 1Centre for Data Analytics and Cognition, La Trobe Business School, La Trobe University, Melbourne, VIC 3086, Australia; i.senadheera@latrobe.edu.au; 2Occupational Therapy, School of Allied Health, Human Services and Sport, La Trobe University, Melbourne, VIC 3086, Australia; blarssen@student.ubc.ca (B.C.L.); y.mak-yuen@latrobe.edu.au (Y.Y.K.M.-Y.); s.steinfort@latrobe.edu.au (S.S.); l.carey@latrobe.edu.au (L.M.C.); 3Department of Physical Therapy, University of British Columbia, Vancouver, BC V6T 1Z3, Canada; 4Neurorehabilitation and Recovery, Florey Institute of Neuroscience and Mental Health, Melbourne, VIC 3086, Australia; 5Department of Occupational Therapy, St. Vincent’s Hospital Melbourne, Fitzroy, VIC 3065, Australia

**Keywords:** stroke, upper extremity, somatosensory disorders, touch, proprioception, haptic object recognition, profiling, unsupervised machine learning

## Abstract

Altered somatosensory function is common among stroke survivors, yet is often poorly characterized. Methods of profiling somatosensation that illustrate the variability in impairment within and across different modalities remain limited. We aimed to characterize post-stroke somatosensation profiles (“fingerprints”) of the upper limb using an unsupervised machine learning cluster analysis to capture hidden relationships between measures of touch, proprioception, and haptic object recognition. Raw data were pooled from six studies where multiple quantitative measures of upper limb somatosensation were collected from stroke survivors (*n* = 207) using the Tactile Discrimination Test (TDT), Wrist Position Sense Test (WPST) and functional Tactile Object Recognition Test (fTORT) on the contralesional and ipsilesional upper limbs. The Growing Self Organizing Map (GSOM) unsupervised machine learning algorithm was used to generate a topology-preserving two-dimensional mapping of the pooled data and then separate it into clusters. Signature profiles of somatosensory impairment across two modalities (TDT and WPST; *n* = 203) and three modalities (TDT, WPST, and fTORT; *n* = 141) were characterized for both hands. Distinct impairment subgroups were identified. The influence of background and clinical variables was also modelled. The study provided evidence of the utility of unsupervised cluster analysis that can profile stroke survivor signatures of somatosensory impairment, which may inform improved diagnosis and characterization of impairment patterns.

## 1. Introduction

Altered somatosensation is common after stroke [1,2,3,4]. In investigations specific to the upper limb, frequency of impaired somatosensation has been shown to be 67% [5]. This can include multiple modalities, such as touch, proprioception, and temperature, with common impairment in texture discrimination, limb position sense, and haptic object recognition [5]. Somatosensory loss is an invisible and often poorly characterized impairment that can have a significant impact on how we interact with our physical environment and participate in daily activities [6,7,8,9]. For example, it contributes to pinch grip deficit after stroke [10], and is associated with functional arm use [11]. Furthermore, presence of somatosensory impairment can negatively affect participation in the number and types of daily living activities, including social and low-demand leisure activities, after stroke [6,12]. Adequate sensation is also reported to be a prerequisite for full motor recovery of the paretic upper limb [13].

Our ability to perceive, recognize, and correctly manipulate objects requires a complex integration of multiple modalities of sensation, including touch, proprioception, and haptic object recognition. However, currently, our ability to profile somatosensory function across multiple modalities, types of information processing, and severity of impairment is limited. When analysing the influences of somatosensation on outcome and functional abilities after stroke, items indexing sensation are often hidden in a global measure of impairment such as the NIH Stroke Scale [14], combined with other measures of sensation to create a single composite score reflecting an average sensory impairment rating [15], or only one modality of sensation is assessed (e.g., light touch). Furthermore, many clinical measures of somatosensation like the Nottingham Sensory Assessment [16], are scored using ordinal ratings that lack sensitivity to the variability in presentation of the assessed impairment (c.f., see [17,18] for examples of quantitative assessments with higher scale resolution). While these scores can be helpful for tracking change if collected at multiple timepoints in the same individual, they fail to illustrate the potentially meaningful variability in impairment both within and across different modalities of sensation, which could be contributing to patient performance on different functional tasks. 

We propose the creation of somatosensory profiles that capture how impairments in different modalities of somatosensation can co-occur within an individual and whether the observed pattern of impairment is potentially representative of a phenotype, signature, or “fingerprint” of somatosensory impairment within a population. Generating profiles has relevance not only in better understanding patterns of somatosensory impairment within and across modalities, but also their impact on function, as different signatures of impairment could differentially impact dexterity and co-ordination of the upper limb, arm use, and/or functional activities. To allow us to characterize the potential patterns or relationships across multiple modalities of somatosensation in a sample of stroke survivors, we applied an unsupervised machine learning approach [19] to characterize the relationship between three different quantitative measures of somatosensory impairment and capacity: the Tactile Discrimination Test (touch [20]), the Wrist Position Sense Test (proprioception [21]), and the functional Tactile Object Recognition Test (haptic object recognition [22]). We selected these measures as they characterize different somatosensory modalities, measures are quantitative with good scale resolution and psychometric characteristics [20,21,22], discriminative impairment is common and persistent after stroke [2,5], and we had access to raw data to pool across multiple data sets.

The co-existence and integration of somatosensory information is complex, involving the simultaneous detection and weighting of relevant stimuli, as well as gating of irrelevant stimuli so that meaningful information can be extracted to inform different cognitive processes, actions, and functional tasks. For example, discriminating where our limbs are in space relative to each other via our proprioceptive senses is essential for the execution of coordinated goal-directed movement [23,24]. Further, our ability to use our sense of touch to discriminate textures can help us recognize objects and inform our decisions about how to safely interact with them, which is frequently identified as an important goal of therapy for patients with somatosensory loss [25]. Each of these senses in isolation is important, and they also work together to support the maintenance of cognitive representations of the body (“body schema”) and performance on more complex functional tasks [22,26,27,28]. For example, our awareness of limb posture paired with information about texture discrimination can inform how we accurately manipulate a key to open a door. A further consideration is that one sensation modality can influence our perception of other somatosensory modalities. For example, using a tactile illusion paradigm, the perception of the location of an illusory tactile stimulation can be manipulated by changing the posture of the hand, suggesting that proprioceptive inputs can influence tactile perception [29]. The contribution and integration of each is important and impacts function.

The relationship between touch and proprioception is made more complex in the context of stroke, where severity of impairment can differ across modalities, and potentially influence each other. These relationships are difficult to unpack for several reasons. First, in clinical research, the specific nature and severity of modalities are often not detailed on an individual level. If multiple modalities of impairment are reported, it is often presented as group averages which fail to capture more nuanced patterns of variability across individuals (e.g., [1,3,4]). Second, sample sizes are often small, making it difficult to identify patterns of impairment that may exist in a given population or sample and prevent us from being able to reveal how modality-specific impairments may interact with each other. Third, summed or composite scores mask the relative presence and severity of contributing impairments for each somatosensory modality that may be impacting an individual’s function. 

Machine learning-based analysis approaches provide an opportunity to explore relationships between variables beyond standard statistical approaches, and to summarize profiles of somatosensory impairment/capacity. Using unsupervised machine learning algorithms, we can use a data-driven approach to reveal previously unseen patterns of relationships in multivariate data sets [19,30]. The advantage of employing unsupervised algorithms for conducting exploratory analyses is that they learn the patterns or rules from the data set with minimal intervention, which mitigates introduction of a priori human bias that can be introduced via labels and classifications [31]. In the context of stroke with impairments that span multiple domains, this approach to pattern analysis lends itself well to understanding if and how the relative severity of impairments may co-occur, or cluster, in a population (e.g., [30]). Unavailability of prior labelled data was a further key reason for using unsupervised learning methods in the current study. Clinically, cluster profiling can be a valuable tool in characterizing the profile of somatosensory impairment for an individual relative to population benchmarks. Furthermore, identifying profiles, or a signature of impairment can motivate future work to consider the interactive nature of impairments when tailoring rehabilitation intervention to the individual and measuring change in somatosensory profiles over time.

Our aim was to generate profiles of somatosensory impairment across three different modalities of upper limb sensation: touch, proprioception, and haptic object recognition. Using unsupervised learning methods and quantitative measures of somatosensation within individuals, we sought to systematically identify within a population of individuals whether scores on a set of outcomes map to distinct phenotypes of somatosensory impairment and capacity. This was achieved using pooled raw data from studies of individuals with stroke with known somatosensory impairment. We included assessment scores from both limbs to capture impairments that may also be present in the ipsilesional hand [32,33]. Hand dominance, and whether the affected hand was the dominant hand, were also included as variables of interest as hand dominance has been associated with asymmetries in somatosensory function in healthy controls [34,35].

## 2. Materials and Methods

### 2.1. Study Design and Pooled Samples

Data were pooled from six studies where survivors of stroke were tested using several quantitative measures of somatosensation. Most were intervention studies where the effectiveness of sensory discrimination training for the recovery of somatosensation in the upper limb was investigated. The studies were: (1) Discriminative validity study [20,21]; (2) Study of the Effectiveness of Neurorehabilitation on Sensation (SENSe), [15], Australian New Zealand Clinical Trials Registry: ACTRN012605000609651; (3) Connecting New Networks for Everyday Contact through Touch (CoNNECT), [36]); ACTRN12613001136796; (4) Network of sites and “up-skilled” therapists to deliver best-practice stroke rehabilitation of the upper limb (SENSe CONNECT: ACTRN12618001389291; (5) In Touch [37]; and (6) additional testing linked with the National Institute of Health (NIH) Toolbox study [38]. Reporting on measures from this pooled sample has also been published elsewhere [39,40].

To assess for relationships across multiple modalities of somatosensation on an individual subject level, only studies that included at least two of the following quantitative, standardized assessments of upper limb somatosensation were eligible for inclusion: TDT (Tactile Discrimination Test (TDT [20]), Wrist Position Sense Test (WPST [21]), functional Tactile Object Recognition Test (fTORT [22]). The TDT [20] and WPST [21] were collected for all six studies, and the fTORT [22] was collected for four of the included studies (SENSe, CoNNECT, In Touch, and SENSe CONNECT). Testing was conducted on both the contralesional (typically affected) and ipsilesional (typically less-affected or unaffected) upper limbs by trained assessors. For studies that had multiple timepoints of assessment, only the first assessment was included in the pooled data set. This baseline assessment was conducted prior to any somatosensory intervention. All participants gave written voluntary informed consent and ethical approval for all studies was granted by the Human ethics committees of Austin Health, La Trobe University and participating institutions/hospitals. The ethical approval for the current project involving the analysis of pooled data was granted by Austin Health HREC/17/Austin/281 and La Trobe University Human Ethics Committees, Melbourne, Victoria, Australia.

Sample size: In cluster analysis, the goal is typically to group similar data points into clusters based on a similarity measure or distance metric. We used GSOM to generate a topology-preserving mapping of the data, which was then separated into clusters using K-means. It is an unsupervised learning technique and does not involve hypothesis testing or the need to estimate effect sizes. Therefore, power calculation is not a standard requirement when performing cluster analysis for recognizing patterns [41]. Nevertheless, the sample size we used in this experiment exceeds the minimum sample size mentioned in the literature [41,42]. 

### 2.2. Participants

Participant inclusion and exclusion criteria and testing conditions were similar across all studies. Participants were recruited from hospitals, rehabilitation centres, and the community, and spanned subacute to chronic phases of recovery. Most were recruited from metropolitan sites in Melbourne, Victoria, Australia. Common inclusion criteria were participants with ischaemic or haemorrhagic stroke, all participants were medically stable, had adequate comprehension of simple instructions, and were able to provide informed consent. Participants presented with impaired touch discrimination, limb position sense, and/or tactile object recognition identified clinically and by standardized tests, with the exception of those from the Discriminative Validity [20,21] and NIH Toolbox [38] studies where participants with and without sensory loss were recruited. Studies differed with respect to criteria related to stroke chronicity (i.e., time since stroke). Participants were eligible for SENSe if they were at least six weeks post-stroke, CoNNECT if they were at least three months post-stroke, and In Touch if they were at least one month post-stroke. The Discriminative Validity and SENSe CoNNECT studies had no restrictions on stroke chronicity. Stroke chronicity data were not collected for the NIH Toolbox study. Stroke survivors were excluded if they were medically unwell, had central nervous system dysfunction other than stroke, diagnosis of peripheral neuropathy, or presence of unilateral spatial neglect. Study exclusion criteria differed with respect to stroke lesion location. Individuals with brain stem infarcts were excluded from CoNNECT and In Touch studies, but not the other studies. Furthermore, CoNNECT and In Touch studies only included participants with first ever stroke and were right hand dominant. There were no such restrictions in the other studies. However, for the purposes of the current study, only those with first ever stroke were included. As CoNNECT and In Touch were neuroimaging studies, participants who did not meet eligibility criteria with respect to MRI safety were not eligible.

### 2.3. Somatosensory Assessments

#### 2.3.1. Tactile Discrimination Test (TDT) 

The TDT is a quantitative assessment of the ability to discriminate precisely defined surface textures (grids) through the sense of touch, using one’s preferred finger tip (either index or middle), with vision occluded [20]. It is a three-alternative, forced-choice assessment where participants have a 33% chance of guessing correctly. This assessment has high retest reliability (r = 0.92) and good discriminative validity [20]. Test procedures for this assessment are published in Carey et al. [20]. All participants in the current study completed the 25-item version of this assessment. A percent maximum area (PMA) raw score ranging from 0 to 100 was calculated for each participant. This is an updated scoring approach [39]. A score less than or equal to 73.1 PMA is associated with impaired performance (95th percentile criterion of abnormality relative to age-matched healthy controls) [40]. 

#### 2.3.2. Wrist Position Sense Test (WPST)

The WPST is a quantitative assessment of an individual’s perceived position of their wrist in the absence of vision [21]. The examiner moves the participant’s wrist to 20 different imposed test positions within a comfortable range of wrist flexion and extension. Test positions span 35 degrees extension and 65 degrees flexion. The participant indicates the perceived joint angle of their wrist using a protractor scale and pointer aligned with the axis of movement at their wrist. An error score is calculated relative to the actual wrist angle for each position. A mean error score is calculated from the absolute value of error at each position. This assessment has high retest reliability (r = 0.88 to 0.92) and good discriminative validity [21]. Mean error scores greater than or equal to 11.3 degrees are considered impaired [39].

#### 2.3.3. Functional Tactile Object Recognition Test (fTORT)

The fTORT is a 14-item quantitative assessment of an individual’s ability to recognize everyday objects through touch. Detailed procedures are presented in Carey et al. [22]. In brief, participants are presented with different objects with different somatosensory attributes. Without vision, they must identify what the object is by indicating the matching object from object sets presented on a poster. Object sets vary according to 7 somatosensory attributes (size, shape, weight, texture, hardness, temperature, and object function), with each attribute sampled twice. Each test item is scored on a scale of 0–3, where 3 is correct identification of the test object (i.e., exact match of object sensory attribute); 2 is object pair (i.e., error in amount of distinctive sensory object attribute); 1 is distractor object (i.e., error in recognition of two or more sensory object attributes); and 0 is incorrect object outside the object set (i.e., error of object type/function and sensory object attribute) [22]. Scores less than or equal to 39.5 are indicative of impaired performance [39]. A summary of all somatosensory assessments and scoring is provided in Table 1.

### 2.4. Data Analysis

Figure 1 illustrates our process by which an unsupervised learning approach based on GSOM was used to generate a topology-preserving mapping of the data, which was then separated into clusters using K-means [43]. We employed this process for two independent analyses. The first included the full sample of participants with TDT and WPST data available (*n* = 203) to investigate the binary relationship between these two modalities in a larger sample. The second included participants with complete TDT, WPST, and fTORT data sets for both hands (*n* = 141) to further profile how touch, proprioception, and haptic object recognition cluster. The selected sample size for each analysis varied according to completeness of the available data for the variables to be analysed.

#### 2.4.1. Quantitative Variables

The total pooled sample included in the study consisted of 207 participants with stroke and somatosensory impairment. The dataset consisted of three discrete measures (1 for each modality of somatosensation: TDT, WPST, and fTORT) × 2 limbs (contralesional and ipsilesional hand) for each participant, which were collected at a single time point. In addition, some categorical variables related to key demographic and background clinical information were available (e.g., age, gender, lesion level, lesion side, lesion type, affected upper limb, dominant hand, is dominant hand affected, and latency—weeks between date of stoke to time of TDT baseline assessment).

#### 2.4.2. Data Pre-Processing

Data pre-processing is an essential step in data analysis since it involves cleaning, transforming, and preparing the data before it is used to train a machine learning model. The goal of data pre-processing is to make the data usable for the clustering algorithm by removing any irrelevant or redundant information, handling missing values, and ensuring that the data is in a suitable format for the chosen algorithm.

Since neither data imputation nor data interpolation yields correct participant clinical scores, we therefore deleted data rows with missing values for clinical scores considered for specific analysis [44]. Treating outliers was not required since no outlier values were noticed in the dataset.

#### 2.4.3. Data Normalization

The pre-processed dataset had clinical scores of different scales. For example, TDT scores range from 0 to 100, WPST’s range is 0–40, and fTORT scores range from 0 to 42 (see Table 1). Since many machine learning algorithms are sensitive to the scale of the input data, it was necessary to normalize data to ensure that each feature contributes equally to the final result. Data normalization involved transforming all data features to lie within the range of 0 and 1. This is done to ensure that the magnitude of the features does not affect the results of the GSOM algorithm. We used the min–max scalar algorithm to normalize clinical scores and used normalized data to construct an input feature vector for the GSOM algorithm [19].

#### 2.4.4. Feature Engineering of Categorical Information

Feature engineering of categorical features involved the transformation into a suitable format for the machine learning algorithm. Categorical variables such as background clinical information (i.e., hand dominance, dominant hand is affected hand) were transformed using one-hot encoding [45]. One-hot encoding involves converting each categorical feature into a set of binary variables, where each binary variable corresponds to a single category. Relevant feature vectors from the results were then used to train the GSOM model for each experiment. 

#### 2.4.5. Growing Self-Organizing Map (GSOM)

A self-organizing map (SOM) is a type of artificial neural network (ANN) used for unsupervised learning-based clustering. A SOM projects high-dimensional input data onto a lower-dimensional map, where similar input patterns are clustered together. The map is self-organizing in that it adapts to represent the structure of the input data over time. The GSOM [19] is an extension of SOMs that allows the map to grow and adapt to new data as it is encountered without the dependence on a pre-defined fixed structure. It can also detect and handle outliers and noise in the data, making it well-suited for unsupervised data exploration, anomaly detection, and data mining applications. 

GSOM was selected as the clustering method for this study since GSOM does not require a pre-defined network architecture which is difficult to identify in an exploratory study such as this study. The GSOM is also flexible and adaptable for small datasets where the data distribution may need to be better defined. This enables unbiased exploration of data patterns and does not require pre-defined labels.

In this study, we used GSOM’s spread factor (SF) hyperparameter value 0.9. The spread factor parameter, which can take values between 0 and 1, controls the node growth in the GSOM allowing the generation of maps with small or large number of nodes. Higher spread factors (larger maps) lead to uncovering more detailed data patterns (including non-prominent sub groupings), while lower spread factors (smaller maps) yield high-level data patterns (only the most prominent groupings).

#### 2.4.6. Identifying Clusters from the GSOM

The GSOM results in a two-dimensional grid of nodes, where each node represents a cluster of similar data patterns. The nodes spatially close to each other on the map are thought to represent clusters. Hence, we could even examine the map and identify areas where the nodes are densely packed, as these areas typically represent clusters of similar patterns.

However, in this study, we used another algorithm (K-means) [43] to derive the partitioning on the GSOM map as a second step. The quality of uncovered clusters were then evaluated with internal cluster validation methods such as silhouette coefficient (SC) [46] and Davies–Bouldin index (DBI) [47], and with statistical methods such as independent sample *t*-tests. Furthermore, two-sample *t*-tests were used to confirm the distinctiveness of the clusters statistically.

## 3. Results

### 3.1. Demographics and Key Background Clinical Information

A summary of participant demographic data and background clinical information is reported in Table 2 and Figure 2.

### 3.2. GSOM Cluster Analysis

The following sections describe GSOM-based clustering to generate a profile of somatosensory impairment across different modalities of upper limb sensation: touch, proprioception, and haptic object recognition. Cluster generation was carried out to help answer the following main questions: (a) How do survivors of stroke cluster according to TDT and WPST scores for the contralesional hand, ipsilesional hand, and both hands? (b) How do survivors of stroke cluster according to TDT, WPST, and fTORT scores for the contralesional hand, ipsilesional hand, and both? (c) Does the hand affected and hand dominance impact the clusters? 

#### 3.2.1. Somatosensory Impairment Profiles across Two Modalities (TDT and WPST)

We examined how survivors of stroke cluster according to TDT and WPST scores for contralesional hand, ipsilesional hand, and both hands. 

The following figures display the result profiles from the clustering experiments across two modalities: TDT and WPST for contralesional hand (Figure 3); ipsilesional hand (Figure 4); and both hands (Figure 5).

Table 3 displays the summary of the result profiles from all three clustering experiments across TDT and WPST.

##### 
Cluster-Based Profiles with TDT and WPST for Contralesional Hand


Altered somatosensation was profiled across TDT and WPST for the contralesional hand and three profiles of individual level data patterns were observed (*n* = 204). Figure 3 and Table 3 (a) display the summary of the profiles.

Profile 1 (25% of analysed participants) presented with no to mild touch impairment and relatively mild to no proprioception impairment for the contralesional hand. Participants of Profile 2 (49%) exhibited relatively severe touch impairment and no to mild proprioception impairment for the contralesional hand. Profile 3 (26%) had participants with moderate to extreme impairment in both touch and proprioception. 

##### 
Cluster-Based Profiles with TDT and WPST for Ipsilesional Hand


Somatosensory impairment was profiled across TDT and WPST for the ipsilesional hand, and three profiles having weak cluster structure were observed at individual-level data patterns (*n* = 203). Figure 4 and Table 3 (b) display the summary of the profiles.

Profile 1 (25% of analysed participants) typically ranged from mild to moderate impairment in both touch and proprioception. Profile 2 participants (40%) mostly had no impairment in both modalities. Profile 3 (35%) exhibited no proprioception impairment mostly for the ipsilesional hand, while touch sensation varied from no to mild impairment levels. 

##### 
Cluster-Based Profiles with TDT and WPST for Both Contralesional and Ipsilesional Hands


Somatosensory impairment was profiled across TDT and WPST for both contralesional and ipsilesional hands and three profiles having weak cluster structure were observed at individual level data patterns (*n* = 203). Figure 5 and Table 3 (c) display the summary of the profiles.

Profile 1 consisted of participants (40%) with no to mild touch impairment (high variance) and no to mild proprioception impairment in the contralesional hand. Their ipsilesional hand exhibited no to mild touch and proprioception impairment. Profile 2 (26%) ranged from moderate to severe touch impairment (high variance) and no to mild proprioception impairment for the contralesional hand, with mild impairment in touch for the ipsilesional hand. Profile 3 (34%) included participants with a range of moderate to severe contralesional impairment for touch and proprioception (high variance) and no to mild impairment in ipsilesional hand in both touch and proprioception.

#### 3.2.2. Somatosensory Impairment Profile Generation across Three Modalities (TDT, WPST and fTORT)

As a further step, we examined how somatosensory impairment based on the TDT, WPST, and fTORT is grouped into clusters, for contralesional, ipsilesional, and both hands, and how the two variables “handedness” and “is the affected hand the dominant hand” impact the clusters. 

The following figures display the result profiles from the clustering experiments across three modalities: TDT, WPST, and fTORT for contralesional hand (Figure 6); and ipsilesional hand (Figure 7).

Table 4 displays the summary of the result profiles from two clustering experiments across three modalities for (a) contralesional hand; and (b) ipsilesional hand.

##### 
Cluster-Based Profiles with TDT, WPST, and fTORT Values for Contralesional Hand


After deleting records with missing values, 144 participants were included in the experiment (*n* = 144). These participants were from the SENSe, In Touch, CoNNECT, and SENSe CONNECT studies, all sensory intervention studies. Three profiles were observed through the cluster analysis. Figure 6 and Table 4 (a) display the summary of the uncovered profiles.

In the analysed sample, 28% of participants were classified into Profile 1, showing a range from moderate to severe touch, proprioception, and haptic object recognition impairment. Profile 2 consisted of 47% of participants, with varying severity from mild to moderate touch impairment (high variance), no to mild proprioception, and mild to moderate haptic object recognition impairment (high variance). Profile 3 consisted of 25% of participants, showing relatively moderate impairment across all modalities. 

##### 
Cluster-Based Profiles with TDT, WPST, and fTORT for Ipsilesional Hand


We profiled somatosensory impairment across TDT, WPST, and fTORT for ipsilesional hand (*n* = 141) and observed four major profiles. Figure 7 and Table 4 (b) display the summary of the profiles.

Profile 1 (23% of the analysed participants) showed mild touch impairment in the ipsilesional hand, but no proprioception and/or haptic object recognition impairment. Profile 2 (21%) had varying levels of mild touch impairment, but no proprioception impairment, with a range from no to mild haptic object recognition impairment. Profile 3 (27%) showed no impairment in touch, proprioception or haptic object recognition for the ipsilesional hand. Profile 4 (29%) had no to mild touch and haptic object recognition impairment, with varying levels from no to moderate proprioception impairment.

Furthermore, we examined how somatosensory impairment based on the TDT, WPST, and fTORT is grouped into clusters, for both contralesional and ipsilesional hands, and how the two variables “handedness” and “is the affected hand the dominant hand” impact the clusters.

The following figures display the result profiles from the clustering experiments across three modalities for both hands (Figure 8); and when clinical background information, handedness, and affected hand’s dominance variables are included in the analysis (Figure 9).

Table 5 displays the summary of the result profiles from two clustering experiments for both hands across (a) three modalities; and (b) three modalities when clinical background information including handedness and affected hand’s dominance variables are included in the analysis.

##### 
Cluster-Based Profiles with TDT, WPST, and fTORT for Both Contralesional and Ipsilesional Hands


Somatosensory impairment was profiled across TDT, WPST, and fTORT for both contralesional and ipsilesional hands as the next step for participants with complete data sets for both hands (*n* = 141). Four major profiles were identified. The summary of profiles is described in Figure 8 and Table 5 (a).

Profile 1 consisted of participants (25%) with no to mild impairment in all three modalities for the contralesional hand. Their ipsilesional hand showed no to mild impaired touch impairment, with no proprioception or haptic object recognition impairment. Profile 2 (26%) ranged from moderate to severe impairment in touch, proprioception, and haptic object recognition for the contralesional hand, with varying levels of no to mild touch impairment (high variance), and no impairment in proprioception or haptic object recognition for the ipsilesional hand. Profile 3 (20%) included participants with a range of moderate contralesional impairment (high variance) and no to mild impaired ipsilesional hands in all three modalities. Participants of Profile 4 (29%) had mild to severe impaired touch, with varying levels from no to moderate impaired proprioception and haptic object recognition for the contralesional hand. Their ipsilesional hand showed no to mild impairment across modalities. 

##### 
Inclusion of Handedness and Affected Hand Dominance Variables into Somatosensory Impairment Profiles Generated Using TDT, WPST, and fTORT for Both Contralesional and Ipsilesional Hands


We included the following parameters in the above analysis of somatosensory assessment data from both hands: (i) dominant hand (DH), (ii) is dominant hand the affected hand (DAH). Three clusters emerged (Figure 9). Profiles are reported in Table 5 (b). 

Profile 1 (36%) comprised participants having mild to moderate impairment across the three modalities for the contralesional hand, and no/mild impairment for the ipsilesional hand for touch only. Ninety percent were right hand dominant, with 32% dominant hand affected. Profile 2 (38%) showed moderate contralesional impairment across modalities for the contralesional hand, with high variability. Again, the ipsilesional hand showed no/mild impairment for touch only. Ninety percent were right hand dominant, with 100% having the dominant hand affected. Participants of Profile 3 (26%) had severe impairment across all modalities for the contralesional hand, again with no/mild impairment for touch only in the ipsilesional hand. All were right hand dominant with only 8% dominant hand affected.

Table 6 shows the cross-cluster analysis between the derived profiles of the above two analyses. All four somatosensory impairment profiles across TDT, WPST, and fTORT for both contralesional and ipsilesional hands (analysis A) were re-organized across three profiles when handedness and affected hand’s dominance variables were entered (analysis B). Participants of Profile 1 in analysis A were distributed among Profiles 1 and 2 of analysis B. Most of Profile 2 of analysis A was re-allocated into Profiles 2 and 3 of analysis B. Participants in Profiles 3 and 4 of analysis A were re-grouped among all three profiles in analysis B.

## 4. Discussion

Signature profiles of somatosensory impairment across modalities of touch discrimination, proprioception, and haptic object recognition were characterized for the contralesional hand and ipsilesional hand of survivors of stroke. Distinct profiles were characterized according to somatosensory modality, relative presence/severity of impairment, and spread/variability of scores. Two-modality (TDT and WPST) and three-modality (TDT, WPST, fTORT) analyses, plus the addition of background clinical information, uncovered distinct profiles for contralesional, ipsilesional, and for combined hands. Three to four distinct profiles (clusters) were identified in each sub-analysis, with relatively similar frequencies of survivors belonging within a profile i.e., ranging from 25–49% per profile (when three profiles emerged), and 20–29% (for four emergent profiles).

### 4.1. Signature Profiles of Somatosensation Post-Stroke

Characterization of somatosensory signatures using unsupervised learning-based cluster analysis has provided new insights into profiles of clinical impairment of sensation and residual capacity, after stroke. Clustering according to two somatosensory modalities (touch discrimination and wrist position sense) highlighted a separation in clusters (profiles) according to presence of impairment in one (usually touch) (Table 3 (a), Profile 2, 49%) or both modalities (Profile 3) for the expected contralesional (typically affected) hand. This observation is consistent with clinical observation but has not previously been identified as a distinct somatosensory impairment signature. Somatosensory profiles for the contralesional hand were also characterized by the relative severity of impairment across one or other modality, for example, with Profile 1 showing unimpaired to mild impairment across both modalities. In the current study, both modalities were quantified and had independently defined criterion of just noticeable impairment and severe impairment, which is not usually the case in clinical practice. Impairment was also present in the ipsilesional hand (typically presumed to be “unaffected”) for touch only or both modalities (40%) across two profiles (Table 3 (b), Profiles 1 and 3), but was usually relatively mild, as previously reported [5]. Profiling that included both hands revealed that those with relatively mild impairment in touch and proprioception for the contralesional hand, usually had little/no impairment in the ipsilesional hand, as might be expected (Profile 1). Surprisingly there was a subgroup who had marked contralesional hand impairment in both modalities but little/no ipsilesional hand impairment (Profile 3). In comparison, Profile 2 was mixed, with marked touch and mild proprioception impairment in the contralesional hand, and ipsilesional hand impairment mostly in the touch modality. Thus, the three profiles that emerged for each analysis revealed distinctive impairment within a modality across hands, across modalities within a hand, or mixed profiles. The addition of assessments from the ipsilesional less-affected hand may be useful in further differentiating those who present with more severe sensory loss.

Somatosensory profiles using the three modalities (touch, limb position sense, and haptic object recognition) emerged based primarily on severity of impairment for this sample who were recruited to sensory intervention trials. Profiles for the contralesional hand were characterized as severe impairment (Profile 1), mild impairment (Profile 2) or moderate impairment (Profile 3) across all modalities. Those with severe impairment in one modality tended to be severely impaired in others. Again, relatively mild ipsilesional hand impairment was evident for three of the four profiles for touch (Profiles 1 and 2), haptic object recognition (Profile 2) or proprioception (Profile 4). When both hands were considered, impairment ranged from no to severe impairment for the contralesional hand, with little or no ipsilesional hand impairment. Although not tested here, the severity-related profiling across modalities is likely impacted by injury to common neural pathways known to be part of a distributed network of brain regions that is responsible for processing somatosensory information [48,49]. Together, identification of these profiles has potential impact not only in relation to the nature and extent of assessment recommended but also the relative impact of multi-modality and multi-hand impairment on sensory rehabilitation training approaches. 

A feature of analyses of the three somatosensory modalities for the contralesional hand was that haptic object recognition impairment, measured with the fTORT, was consistently a distinguishing feature between profiles. Further, while there was evidence of distinction in presence and severity of impairment for the two modality analyses, the severity of impairment was more consistent across modalities for the three modality analyses when haptic object recognition was included. Haptic object recognition is dependent on the ability to integrate both touch and proprioceptive information and relies on cognitive resources such as working memory and attention to form a percept of the object informed by shape and texture [50]. Underpinning these functions are common neural pathways [26] that culminate in shared neural correlates in the parietal operculum (e.g., secondary somatosensory cortex) that are associated with processing all three somatosensory modalities [50,51,52]. Hence, it is not surprising that we observed that the profiles that exhibited severe impairments in touch and proprioception also had severe impairments in haptic object recognition. These data suggest that inclusion of measures of haptic object recognition may be useful in distinguishing subgroups of somatosensory impairment and capacity after stroke.

We also observed how accounting for hand dominance and whether the more affected hand was the dominant hand changed subgroups of impairment. With the addition of these two parameters, the four profiles (Table 5 (a)) redistributed into three profiles (Table 5 (b)). Wide variability in severity was observed within and across modalities for the contralesional hand profiles, with little/no ipsilesional hand impairment, except of touch, across profiles. Profile 2, where the dominant (usually right) hand was affected for all participants, showed the greatest variation in haptic object recognition scores, ranging from relatively mild to moderate and severe impairment. Those in Profile 2 also showed wide variability in proprioception for the contralesional hand. In comparison, proprioception was consistently and most severely affected in the contralesional limb for Profile 3 participants, where only 8% had the dominant right hand affected (i.e., the left non-dominant hand was impaired for the vast majority in this subgroup). Our finding of most severe proprioceptive impairment in those with right hemisphere lesions is consistent with previous evidence that proprioception function for both arms may be partially lateralized to the right hemisphere, specifically the right supramarginal gyrus [53]. Others have also identified that lesions to the supramarginal gyrus, among other regions of interest (arcuate fasciculus and Heschl’s gyrus), are associated with persistent proprioceptive deficits in the contralesional upper limb post-stroke, with larger proportions of damage being associated with more severe deficits [51]. Marked impairment in touch discrimination for the left affected hand in this group is also consistent with evidence of bilateral activation of the supramarginal gyrus in response to unilateral tactile stimulation [52]. The Profile 3 subgroup also showed marked impairment in contralesional hand haptic object recognition, presence of ipilesional hand touch impairment, and individuals with impairment in the ipsilesional hand for proprioception and haptic object recognition. Although not tested here, future work using techniques like lesion–symptom mapping [1,51,54] could be used to investigate the relationships between these profiles and lesion location. 

To our knowledge, our study is the first to investigate how multiple modalities of somatosensory impairments cluster at an individual subject level with a large enough sample to identify potential sub-groupings or clusters of impairment, referred to here as profiles. For example, we have previously reported on the frequency of stroke patients presenting with discriminative sensory loss of the hand across touch and proprioception in the post-acute rehabilitation phase, but this was only at the group level [5]. While we were able to also report on the numbers with impairment in one or both modalities, and look at the relationship between severity across modalities and hands at a group level, we have not previously been able to quantitatively characterize individual participant profiles. Other studies have looked at how measures of somatosensation may differ between subgroups of stroke survivors. In a study comparing groups of stroke survivors with and without post-stroke shoulder pain, different proportions of individuals with impaired touch and proprioceptive impairments are reported; however, it does not identify the frequency of individuals where both senses were impaired, nor does it report on impairment severity [55]. 

In the present study, we have demonstrated that individuals can present with impairments across multiple modalities of somatosensation not only within the same limb, but also across contralesional and ipsilesional limbs. Individuals who experience patterns of impairment across multiple modalities will likely have unique challenges in interacting with their environment compared to those who are impaired in a single somatosensory domain and may have different recovery trajectories. Generating profiles has relevance for investigating the impact of patterns of impairment on function, as different signatures of impairment could differentially impact arm use, participation, and recovery trajectories. Now that somatosensory signatures have been identified, it would be of value to investigate the relationship between such profiles and functional outcomes such as arm use and participation in daily activities.

We also demonstrated that there is value in including assessments of both hands. Through the addition of ipsilesional hand scores we were able to identify profiles of individuals that may have more widespread impairment. Indeed, we found a somatosensory signature where individuals with severe impairment in their contralesional hand often experienced mild impairment in the tactile discrimination test with their less-affected hand and sometimes impaired position sense (three modality analysis, Profile 4, Table 5 (a)). Surprisingly, however, there was also a subgroup in the two modality analysis who did not show ipsilesional hand impairment despite moderate to severe impairment of touch and proprioception in the contralesional limb (two modality analysis, Profile 3, Table 3 (c)). Impairment of the ipsilesional upper limb in addition to the contralesional upper limb has previously been reported [5,33,56] and is consistent with evidence of tract-specific changes associated with touch impairment of the ipsilesional hand [32]. Our current finding suggests the value of further investigation of factors associated with ipsilesional hand impairment, including interruption to specific brain networks. For now, our findings support a rationale to assess for deficits in the ipsilesional hand, especially for those who present with severe impairment in their contralesional affected hand, and consideration of inclusion of intervention approach that target rehabilitation of both hands.

In summary, characterizing profiles of somatosensory impairment across multiple modalities for an individual is indicated, given evidence of distinctive patterns in the presence and severity of impairment across modalities. Further, assessment of the ipsilesional hand is recommended, especially for those with severe impairment on the contralesional limb. By examining the clustering solution, researchers can gain insight into the complex relationships between different somatosensory impairments and potentially highlight modalities that are most impaired and help inform tailoring interventions and treatment planning to address the specific deficits of subgroups and individuals. 

### 4.2. Advantages and Clinical Utility of Using Unsupervised Machine Learning to Profile Somatosensory Impairment Patterns

Generating signature profiles of clinical outcomes for an individual brings us one step closer to personalized medicine or personalized rehabilitation. The application of data-driven multifactorial analysis in health care, termed “rehabilomics” [57], is emerging as an area of interest in the field of rehabilitation. Profiling or evaluating patients based on measures related to demographics, measures of clinical status, and measures of recovery status can all be used to develop models that have the potential to predict outcome and personalize health care (for a recent review, see [57]). Our focus was on somatosensation, which is multi-modal in nature, thus introducing complexity to understanding the expression of impairment of this function in an individual. We used three standardized measures of somatosensation, which are sensitive to a range of impairments and variability in performance, have well-documented psychometric properties, and have clear criteria of abnormality and extreme impairment [20,21,22,39,40]. 

We used Growing Self-Organizing Maps (GSOM) [19] to capture the individuality of somatosensory capacity across multiple modalities, also taking into account the hand affected. Generating signature profiles involves multiple variables, which can be complicated to summarize or aggregate. Application of artificial intelligence (AI) and deep learning to this multi-modal data allowed generation of new insights, as visualized in our profile graphs and plots. A strength of using unsupervised clustering approaches like GSOM, is that it is well suited to managing multidimensional data sets and has the flexibility to find non-linear relationships in data compared to other statistical analysis approaches which require assumptions on data distribution [58]. To date, common machine learning-based approaches that have been applied in stroke rehabilitation are primarily regression-based methods [57]. Unsupervised learning cluster analysis can be a powerful tool for clinicians to better understand and profile stroke survivor populations. Cluster analysis can identify novel associations and patterns among different clinical variables while reducing the opportunity for subjectivity in interpreting outcomes. The GSOM approach has the further advantage that it enables unbiased exploration of data patterns and does not require pre-defined labels in the dataset. It is recognized that the implementation of machine learning approaches in rehabilitation is still an emerging field; thus, further work is needed to not only determine what algorithms and parameters are appropriate for patient profiling and predicting outcome in stroke populations but also to validate the predictive models on larger samples of new data [57]. 

### 4.3. Future Directions

Our current study has mapped distinct profiles of somatosensory impairment across multiple modalities at a single timepoint. While this provides new insights into how impairments co-occur for different clusters of individuals, it does not provide us with a robust indicator of recovery over time. Characterizing impairment profiles at an individual level opens the potential to better characterize sensory recovery trajectories, which is currently limited. In a small sample of nine acute stroke participants, Winward and colleagues [59] found that severity of touch and proprioception impairment varied within individuals at a single timepoint and the recovery trajectory of different modalities varied greatly within and between individuals. For example, despite starting with the same magnitude of severity, one participant demonstrated evidence of partial recovery of proprioception, yet no change in touch detection at one month post-stroke, while another participant who started with similar severe levels of touch impairment, yet milder proprioception impairments, showed a marked recovery in their sense of touch [59]. Although a small and variable sample, these data speak to the importance of indexing not only the presence of impairment across different somatosensory modalities, but also the magnitude of impairment severity within each modality, consistent with our somatosensory profiles. The potential now exists to extend our profile analysis over time, identifying subgroups of individuals with unique combinations or patterns of impairment severity across modalities and over time. 

Future work should map how clusters of impairment may change longitudinally along key timepoints of the stroke recovery continuum [60] and after intervention. This would allow us to better understand how somatosensory impairments change as a part of neurobiological recovery processes, as well as be used to inform clinical predictors of an individual’s capacity for recovery and/or ability to benefit from therapy. Characterization of signatures of impairment and recovery would also permit interpretation of an individual’s recovery relative to such benchmarks. A batch implementation of GSOM may be used to capture patterns in such trajectory data [61]. With the ability to preserve topological relationships in data, GSOM is well placed to represent trajectory patterns. In a clinical setting, this may enable development of automated deviation alerts in the future. 

This approach does not need to be restricted to profiling of somatosensory impairment. A strength of unsupervised clustering techniques is the capacity to explore potentially meaningful relationships with large multivariate data sets. With respect to stroke recovery, many large data repositories including national stroke registries have been developed to facilitate meta-analyses of large pooled samples of data to advance scientific discovery and stroke care [62,63]. Stroke recovery is complex and involves many interacting systems that may differentially impact recovery given the relative severity of impairments, together with presence of different biopsychosocial factors [64,65]. Inclusion of data related to severity of specific motor impairments and spasticity, could expand meaningful profiles of impairment across somatosensory and motor domains after stroke. Accounting for specific lesion location and other indices from neuroimaging data (if available) can also shed light on the importance of critical anatomy associated with impairments and impairment profiles. Key demographics and background clinical information can also be included to extend the current analysis and predictive models. Analysis could be further enriched by introducing other forms of data such as video and wearable sensors, and using multi-modal fusion to cluster behavioural patterns [66]. To account for all these variables in a single statistical model is not feasible [67]. However, application of analyses approaches like GSOM can serve as a starting point to facilitate discovery of relationships between key parameters or unseen common latent variables that better describe and predict recovery in a stroke population. Further, the GSOM algorithm is scalable to capture patterns from large volumes of sensor data and from larger cohorts and more frequent readings [67].

The somatosensory impairment profiles characterized were based on quantitative measures of somatosensory discrimination across tactile (TDT) and proprioceptive (WPST) modalities, and haptic object recognition. As such, the profiles that emerged were linked to these measures of impairment and to the upper limb. They were also defined in stroke populations that span subacute to rehabilitation and chronic phases, providing limits to the generalizability of our findings. Nevertheless, such impairment is common in these populations [1,2,3,4,5], and characterization of patterns of impairment has significance not only in relation to better understanding the nature of impairment, but also providing a means by which specific profiles (phenotypes) might be associated with certain functional outcomes. 

### 4.4. Limitations

Our current study focused on somatosensory variables and creating a sensory signature with a relatively restricted number of variables. The GSOM methodology is capable of handling large volumes of data [61], but this feature was not required for the current analysis. Other factors such as age, presence of motor impairment, time post-stroke, etc., could also influence the profiling of impairment. While these variables were available for most participants, we chose not to include them in the current analysis which was focused on characterizing somatosensory profiles that can be interpreted meaningfully and simply in clinical contexts. Given the demonstrated value of the GSOM approach, the potential now exists to further interrogate the data to discover other profiles that include a wider range of variables. The study could also be further extended by generating GSOMs of varying level of spread and detail in clustering by changing the spread factor parameter between 0 and 1 for each data set to achieve hierarchical clustering. Such further experimentation will provide a more comprehensive understanding of how super clusters (parent, low spread factor) break up into sub-clusters (high spread factor). We are aware that to date these methods may not be as readily accessible and interpretable by clinicians and will likely require support from collaborators for guidance in interpretation. For now, such analyses are opening the door to new insights and new ways of meaningfully capturing the complexity of functions such as somatosensation and how they might be differentially impacted by brain injury such as stroke.

## 5. Conclusions

To our knowledge this is the first study to apply an unsupervised machine learning approach to profile somatosensory impairments post-stroke. Signature profiles were created using data from touch, proprioception, and/or haptic object recognition modalities; contralesional and/or ipsilesional hands; and hand dominance. Through the inclusion of quantitative assessments for both the affected (contralesional) and less-affected (ipsilesional) hand of stroke survivors, we identified distinct profiles (or “fingerprints”) of somatosensation that differed with respect to patterns of impairment presence and severity. A noteworthy finding is that individuals with severe impairment across multiple modalities in the affected hand can be further divided into two distinct subgroups with and without impairment in the less-affected (ipsilesional) hand. These two subtypes of severe impairment may have different challenges with functional tasks which could be an important consideration for tailoring rehabilitation therapies. The feasibility and value of using unsupervised learning-based cluster analysis to profile signatures of somatosensory impairment after stroke was demonstrated and may inform improved characterization of both somatosensory impairment and capacity for targeted rehabilitation and recovery.

## Figures and Tables

**Figure 1 brainsci-13-01253-f001:**
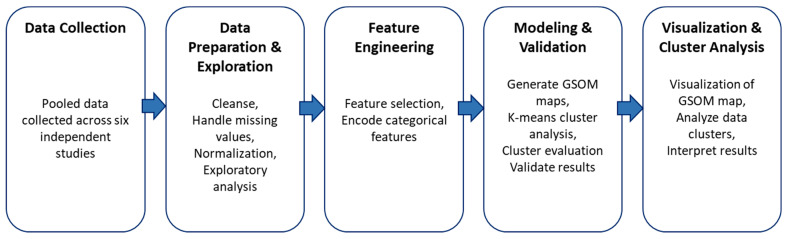
The data analysis process followed to discover patterns from the data.

**Figure 2 brainsci-13-01253-f002:**
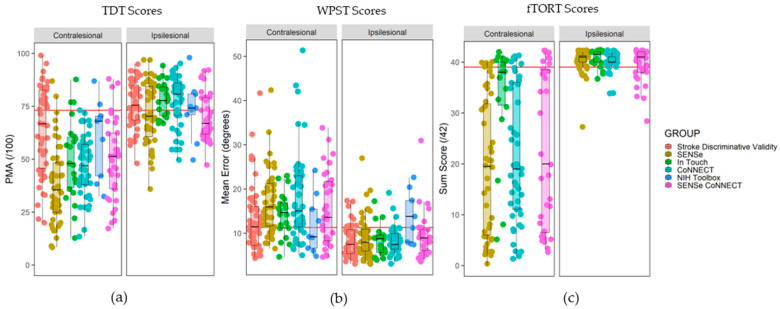
Box plots of somatosensory assessment scores disaggregated by study and hand assessed. Participant scores are represented by individual data points, box plots present the study median score and interquartile range. Separate panels for each hand assessed. The red horizontal line represents the criterion of abnormality for each assessment. (**a**) TDT scores (criterion of abnormality = 73.1 PMA), with lower scores indexing greater impairment. (**b**) WPST scores (criterion of abnormality = 11.3°), with higher scores indexing greater impairment. (**c**) fTORT summed scores out of 42 (criterion of abnormality = 39.5 points), with lower scores indexing greater impairment.

**Figure 3 brainsci-13-01253-f003:**
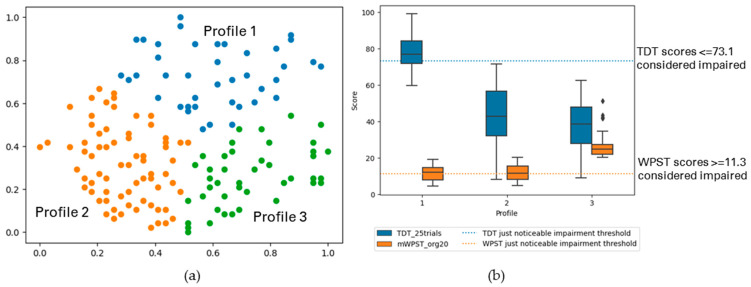
(**a**) The GSOM plot displaying three clusters (spread factor, SF = 0.9). (**b**) Box plots of TDT and WPST scores for contralesional hand disaggregated by each cluster profile. Participants (*n* = 204) are represented in three profiles, box plots present the study median and interquartile range (where diamonds indicate scores beyond 1.5 times interquartile range value).

**Figure 4 brainsci-13-01253-f004:**
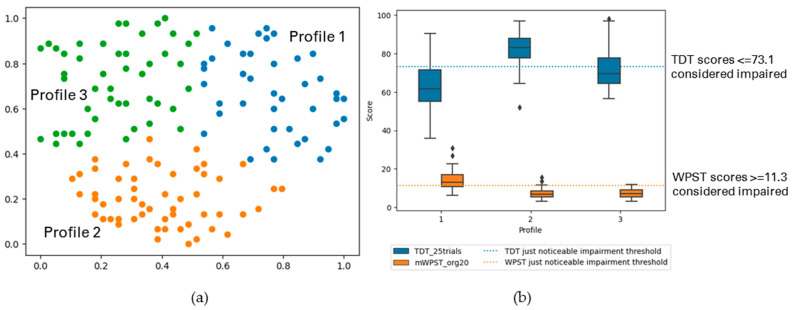
(**a**) The GSOM plot displaying three clusters (spread factor, SF = 0.9). (**b**) Box plots of TDT and WPST scores for ipsilesional hand disaggregated by each cluster profile. A total of 203 participants are represented in three profiles (*n* = 203), box plots present the study median and interquartile range (where diamonds indicate scores beyond 1.5 times interquartile range value).

**Figure 5 brainsci-13-01253-f005:**
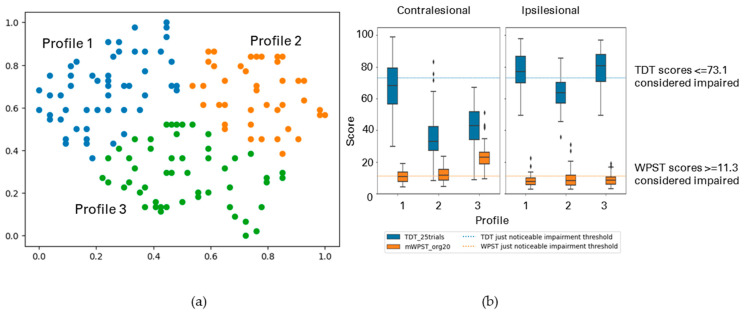
(**a**) The GSOM plot displaying three clusters (spread factor, SF = 0.9). (**b**) Box plots of TDT and WPST scores for both contralesional and ipsilesional hands disaggregated by each cluster profile. A total of 203 participants are represented in three profiles (*n* = 203), box plots present the study median and interquartile range (where diamonds indicate scores beyond 1.5 times interquartile range value).

**Figure 6 brainsci-13-01253-f006:**
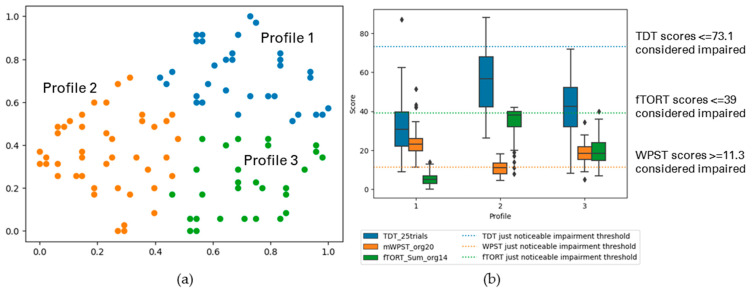
(**a**) The GSOM plot displaying three clusters (spread factor, SF = 0.9). (**b**) Box plots of TDT, WPST, and fTORT scores for contralesional limb disaggregated by each cluster profile. A total of 144 participants are represented in three profiles (*n* = 144), box plots present the study median and interquartile range (where diamonds indicate scores beyond 1.5 times interquartile range value).

**Figure 7 brainsci-13-01253-f007:**
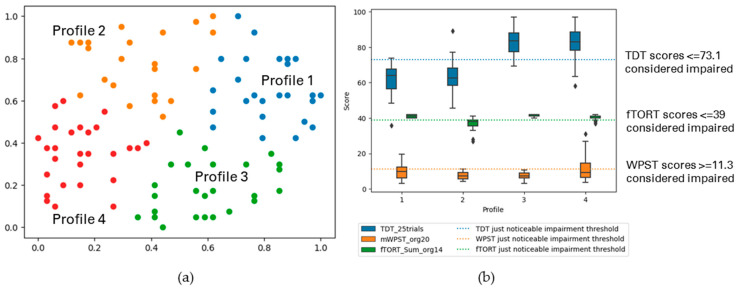
(**a**) The GSOM plot displaying four clusters (spread factor, SF = 0.9). (**b**) Box plots of TDT, WPST, and fTORT scores for ipsilesional limb disaggregated by each cluster profile. A total of 141 participants are represented in three profiles (*n* = 141), box plots present the study median and interquartile range (where diamonds indicate scores beyond 1.5 times interquartile range value).

**Figure 8 brainsci-13-01253-f008:**
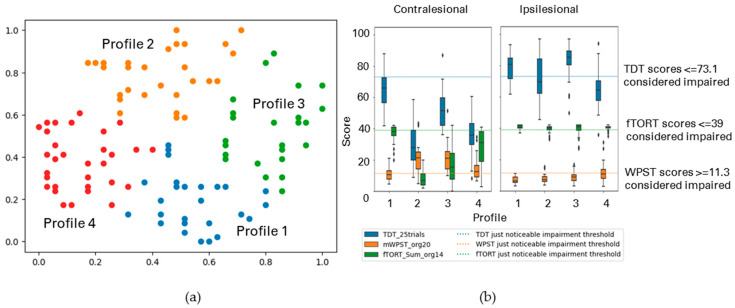
(**a**) The GSOM plot displaying four clusters (spread factor, SF = 0.9). (**b**) Box plots of TDT, WPST, and fTORT scores for both contralesional and ipsilesional hands disaggregated by each cluster profile. A total of 141 participants are represented across three profiles, box plots present the study median and interquartile range (where diamonds indicate scores beyond 1.5 times interquartile range value).

**Figure 9 brainsci-13-01253-f009:**
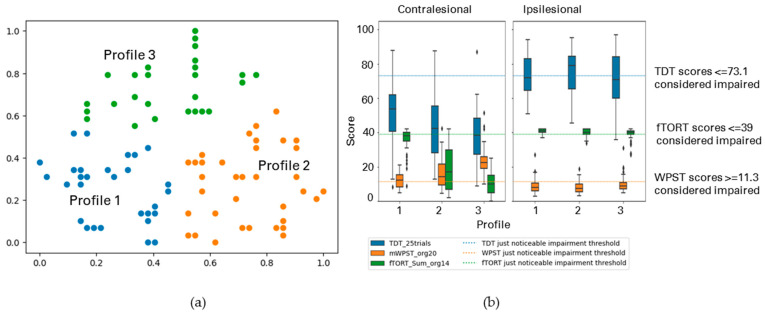
(**a**) The GSOM plot displaying three clusters (spread factor, SF = 0.9). (**b**) Box plots of TDT, WPST, and fTORT scores for both contralesional and ipsilesional hands disaggregated by each cluster profile when handedness and affected hand’s dominance variables are included in the analysis. A total of 141 participants are represented in three profiles (*n* = 141), box plots present the study median and interquartile range (where diamonds indicate scores beyond 1.5 times interquartile range value).

**Table 1 brainsci-13-01253-t001:** Summary of somatosensory assessments TDT, WPST and fTORT.

	Tactile Discrimination Test (TDT)	Wrist Position Sense Test (WPST)	Functional Tactile Object Recognition Test (fTORT)
Assessment score description	Percent Maximum Area under the psychometric function.	Average absolute error (degrees)	fTORT sum score: sum of ordinal scale values across 14 test items
Score range	0–100	0–40	0–42
Interpretation	Higher number = better performance	Higher number = worse performance	Higher number = better performance
Impairment threshold for just noticeable impairment (pooled data) [39,40]	73.1 percent maximum area (PMA)	11.3 degrees average absolute error	39.5 sum score
Impairment threshold for extreme impairment (pooled data) [39]	Values at or near 33.3 PMA and below	Values at or near 36 degrees average absolute error and above	Values at or near 1.5 sum score and below

**Table 2 brainsci-13-01253-t002:** Participant demographic and background clinical information for the total pooled sample (*n* = 207) and disaggregated by study.

	Total	Stroke Discriminative Validity	SENSe	In Touch	Connect	NIH Toolbox	Sense Connect
Sample Size	207	50	46	22	45	9	35
Mean Age (SD)	56.3 (14.5)	52.0 (14.4)	61.3 (11.9)	59.7 (15.1)	52.8 (14.1)	65.2 (12.6)	56.1 (15.7)
Sex: M/F	144/63	36/14	33/13	14/8	32/13	6/3	23/12
Mean Time Since Stroke: weeks (SD)	82.7 (145.5)	13.0 (21.2)	90.9 (124.2)	6 (5.7)	78.3 (92.6)	NA	205.3 (243.5)
Lesioned Hemisphere: R/L/Both	95/108/4	21/28/1	20/26/0	8/14/0	20/23/2	5/4/0	21/13/1
Lesion Level: Cortical/Subcortical/ Both/Unknown	78/65/ 25/39	15/9/ 9/17	13/18/ 10/5	8/13/ 1/0	26/15/ 4/0	0/0/ 0/9	16/10/ 1/8
Hand Dominance: R/L/Unknown	178/25/4	40/6/4	42/4/0	21/1/0	45/0/0	0/9/0	30/5/0
Dominant Hand is Affected Hand: n (%)	101 (48.8%)	21 (42%)	26 (56.5%)	13 (59.1%)	23 (51.1%)	5 (55.6%)	13 (37.1%)

**Table 3 brainsci-13-01253-t003:** Characterization of impairment profiles disaggregated for each impairment profile across TDT and WPST for: (a) contralesional hand; (b) ipsilesional hand; and (c) both hands (*n* = 204).

(a) Impairment profiles across TDT and WPST for contralesional hand	Profile 1	Profile 2	Profile 3
Label	Unimpaired to mild impairment—both modalities	Marked impairment—touch	Marked impairment—both modalities
Sample size (%)	50 (25%)	101 (49%)	53 (26%)
TDT PMA: Mean (SD)	77.9 (8.9)	44.1 (15.0)	37.9 (13.4)
WPST average error: Mean (SD)	11.5 (4.1)	11.9 (4.2)	26.7 (6.9)
**(b) Impairment profiles across TDT and WPST for ipsilesional hand**	**Profile 1**	**Profile 2**	**Profile 3**
Label	Mild-moderate impairment—both modalities	Unimpaired—both modalities	Unimpaired to mild impairment—touch only.
Sample size (%)	50 (25%)	82 (40%)	71 (35%)
TDT PMA: Mean (SD)	64.1 (12.4)	82.1 (7.6)	72.5 (11.0)
WPST average error: Mean (SD)	13.9 (4.8)	7.1 (2.5)	7.4 (2.4)
**(c) Impairment profiles across TDT and WPST for both contralesional and ipsilesional hands**	**Profile 1**	**Profile 2**	**Profile 3**
Label	Mild-no touch and proprioception impairment—contralesional. Mild-no impairment—ipsilesional.	Marked-moderate touch impairment—contralesional. Mild touch impairment—ipsilesional.	Marked touch and proprioception impairment—contralesional only
Sample size (%)	82 (40%)	52 (26%)	69 (34%)
TDT PMA: Mean (SD): Contralesional/Ipsilesional	67.8 (15.1)/77.1 (11.0)	36.2 (15.6)/64.0 (10.8)	41.8 (14.2)/78.7 (10.7)
WPST average error: Mean (SD): Contralesional/Ipsilesional	10.8 (3.6)/8.4 (3.3)	12.3 (5.1)/9.8 (5.9)	23.8 (8.1)/8.9 (3.8)

**Table 4 brainsci-13-01253-t004:** Characterization of impairment profiles disaggregated for each impairment profile across TDT, WPST, and fTORT for (a) contralesional hand (*n* = 144) and (b) ipsilesional hand (*n* = 141).

(a) Impairment profiles across TDT, WPST, and fTORT for contralesional hand	Profile 1	Profile 2		Profile 3
Label	Severe-moderate impairment—all modalities.	Mild impairment—proprioception and object recognition. Mild-moderate—touch.		Moderate impairment—all modalities.
Sample size (%)	41 (28%)	67 (47%)		36 (25%)
TDT PMA: Mean (SD)	33.3 (15.3)	55.2 (16.2)		40.8 (15.5)
WPST average error: Mean (SD)	24.6 (8.7)	10.9 (3.3)		18.5 (5.9)
fTORT summed score: Mean (SD)	5.5 (3.1)	34.4 (8.3)		19.9 (8.0)
**(b) Impairment profiles across TDT, WPST, and fTORT for ipsilesional hand**	**Profile 1**	**Profile 2**	**Profile 3**	**Profile 4**
Label	Mild impairment—touch only.	Mild impairment—touch and object recognition.	No impairment—all modalities.	Mild and variable proprioception impairment.
Sample size (%)	32 (23%)	30 (21%)	38 (27%)	41 (29%)
TDT PMA: Mean (SD)	61.9 (8.6)	63.1 (8.9)	82.2 (7.3)	82.3 (8.8)
WPST average error: Mean (SD)	9.6 (4.0)	7.6 (2.3)	7.2 (2.0)	10.8 (6.0)
fTORT summed score: Mean (SD)	41 (0.8)	36.9 (3.3)	41.6 (0.5)	40.1 (1.2)

**Table 5 brainsci-13-01253-t005:** Characterization of impairment profiles disaggregated for each impairment profile for both contralesional and ipsilesional hands across (a) TDT, WPST, and fTORT; and (b) TDT, WPST, and fTORT when clinical background information including handedness and affected hand’s dominance variables are included in the analysis. (*n* = 141).

(a) Impairment profiles across TDT, WPST, and fTORT for both contralesional and ipsilesional hands	Profile 1	Profile 2	Profile 3	Profile 4
Label	No to mild impairment, all modalities— contralesional. Unimpaired—ipsilesional.	Marked impairment, all modalities— contralesional. No to mild touch impairment— ipsilesional.	Moderate impairment, all modalities— contralesional. Unimpaired ipsilesional.	Mild to marked impairment (high variability)—contralesional. No to mild impairment—ipsilesional.
Sample size (%)	35 (25%)	37 (26%)	28 (20%)	41 (29%)
TDT PMA: Mean (SD): Contralesional/Ipsilesional	65.5 (11.7)/ 79.0 (8.7)	30.5 (12.2)/ 71.6 (12.6)	52.6 (12.0)/ 82.5 (11.1)	36.8 (12.9)/ 64.5 (11.1)
WPST average error: Mean (SD): Contralesional/Ipsilesional	10.8 (4.2)/ 6.8 (2.1)	21.4 (8.4)/ 7.6 (2.7)	21.3 (9.0)/ 9.0 (3.3)	14.2 (6.5)/ 11.7 (5.7)
fTORT summed score: Mean (SD): Contralesional/Ipsilesional	37.1 (5.2)/ 40.7 (1.3)	8.4 (5.4)/ 39.6 (2.3)	17.1 (11.4)/ 39.3 (4.1)	27.4 (12.0)/ 40.4 (1.6)
**(b) Impairment profiles across TDT, WPST, and fTORT for both contralesional and ipsilesional hands when handedness and affected hand’s dominance variables are included in the analysis**	**Profile 1**		**Profile 2**	**Profile 3**
Label	Mild-moderate impairment, all modalities—contralesional. No to mild touch impairment— ipsilesional		Moderate and variable impairment, all modalities—contralesional. No to mild touch impairment— ipsilesional.	Marked impairment, all modalities— contralesional. No to mild touch impairment—ipsilesional.
Sample size (%)	51 (36%)		53 (38%)	37 (26%)
TDT PMA: Mean (SD): Contralesional/Ipsilesional	50.9 (17.3)/ 73.8 (12.1)		44.7 (20.0)/ 75.2 (11.9)	39.0 (15.6)/ 71.0 (15.2)
WPST average error: Mean (SD): Contralesional/Ipsilesional	12.1 (4.0)/ 8.8 (4.3)		16.2 (7.9)/ 7.9 (3.2)	23.5 (9.2)/ 10.4 (5.1)
fTORT summed score: Mean (SD): Contralesional/Ipsilesional	36.0 (6.9)/ 40.8 (1.4)		18.5 (12.8)/ 40.1 (1.9)	10.6 (6.5)/ 39.2 (3.8)
Hand Dominance: R/L	46/5		48/5	37/0
Dominant Hand is Affected Hand: n (% of sample size)	17 (32%)		53 (100%)	3 (8%)

**Table 6 brainsci-13-01253-t006:** Cross-cluster analysis for analysis A (Somatosensory impairment profiles across TDT, WPST, and fTORT for both contralesional and ipsilesional hands) and analysis B (Somatosensory impairment profiles across TDT, WPST, and fTORT for both contralesional and ipsilesional hands inclusive of handedness and dominant hand affected) (*n* = 141).

			Analysis A		
		Profile 1	Profile 2	Profile 3	Profile 4
	**Profile 1**	22	1	14	24
**Analysis B**	**Profile 2**	13	20	10	10
	**Profile 3**	0	16	14	7

## Data Availability

Due to the personal nature of the data and original ethics approval, the data will not be made available broadly. De-identified data may be made available for related research and analysis by the research group and collaborators with additional ethics approval.

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
