# Peer review of "Profiling Somatosensory Impairment after Stroke: Characterizing Common “Fingerprints” of Impairment Using Unsupervised Machine Learning-Based Cluster Analysis of Quantitative Measures of the Upper Limb"

_brainsci, 2023, doi:10.3390/brainsci13091253_

Round 1
Reviewer 1 Report
Dear authors and editors,
Thank you very much for the opportunity to read and review this interesting article. Lots of work supports it, but changes are needed.
Indicating the type of study in the title facilitates and increases the interest of the reader.
In the abstract section improve the introduction. Relevance of the topic. Only upper limb?
Clarify the aim of this study.
Why did the authors choose those 6 articles and not others?
I have doubts about the methodology. The abstract should attract the attention and interest of the reader.
The method seems biased so the conclusions are very ambitious
Use MEsH as keywords.
The introduction is interesting and I detect an enormous amount of effort in the text. However, it could be improved. As I mentioned above, the introduction should highlight the importance of exploring somatosensory alterations in people with stroke. But, why is it relevant? Does it affect their activities of daily living? to their quality of life? to their participation? to their social relationships? Add more information about theses aspects.
Somatosensory alterations affect the dexterity or coordination of the upper limbs?
Is the presence of spasticity related to somatosensory alterations? I suggest adding more information on the relevance of generating profiles of somatosensory alterations. Information related to the difficulties and needs that our patients have.
In relation to Materials and Methods
Have authors used any guides or reference documents to prepare this article? Please include that information.
Please, add information about Ethical committee
I have doubts about the choice and number of items. I do not get it. More information is necessary in this section to clarify.
The selected tests are of interest and relevance, but the authors need to justify their selection. I would like to ask them if monofilaments (Semmes- Weinstein) would be of interest in detecting somatosensory abnormalities.
I suggest that the authors clarify the origin of the participants, the calculation of the sample size...
I have severe doubts and concerns about the methodology that affect the results.
The study presents an excess of tables that need to be grouped.
The discussion is interesting, but implementing all of the above suggestions will require a modification.
Author Response
Please see the attachment, pages 1-6. Reviewer 1 comments are highlighted in yellow.

Reviewer 2 Report
The study provides the data on the varied somatosensory dysfunction that may occur in post stroke patients. The presence of sensory dysfunction impairs ADL and increases the requirement for assistance. Hence the study adds value by describing the signature pattern involves and also assesses the functional component through functional tactile object recognition test. The article is scientifically rich and well written. However, there are few corrections that are suggested. kindly address the punctuation error in line 48, page 2. Consider removing the bold font in the highlighted column in table 2. The author has summarised in table 1 page 5, the various threshold for considering just noticeable impairment and extreme impairment. However, it would add value if the authors elaborate more on the threshold for considering moderate impairment that the authors have used to describe the impairments subsequently in grouping the patients. Kindly address the spelling of impairment in table 8. Kindly address the spelling error in line 628 in page 19. Can the author kindly verify reference 16. The addition of age and gender wise distribution of data will add value to the study and can be provided. Can the authors also explain if the dominance of the individual will play a confounding role? And if the selection of the data of patients from the 6 studies limited to one region?
The quality of English is well written.
Author Response
Please see the attachment, pages 6-8. Reviewer 2 comments are highlighted in yellow.
